# PUPPETMASTER: SCALING INTERACTIVE VIDEO GENERATION AS A MOTION PRIOR FOR PART-LEVEL DYNAMICS

## ABSTRACT

We present PuppetMaster, a video generator that understands *part-level* object dynamics. Given an image of an object and a number of drags defining the desired trajectory of selected points of the object, PuppetMaster synthesizes a video where the object moves according to the specified drags in a physically plausible manner. PuppetMaster is obtained by fine-tuning an off-the-shelf video diffusion model, extended with a new component that encodes the input drags. PuppetMaster also introduces *all-to-first* attention, a replacement for the common spatial attention module, which removes artifacts that arise from fine-tuning a video generator out-of-domain and significantly improves the quality of the synthesized videos. PuppetMaster is learned from Objaverse-Animation-HQ, a new dataset of curated *part-level* motion clips obtained by rendering synthetic 3D animations. We propose strategies to automatically filter out sub-optimal animations and augment the synthetic renderings with meaningful drags. By using this data, PuppetMaster learns to generate part-level motions, unlike other motion-conditioned video generators that mostly move the object as a whole. PuppetMaster generalizes well to real images, outperforming existing methods in real-world benchmarks in a *zero-shot* manner. We refer the reader to the supplementary material for video visualizations.

## 1 INTRODUCTION

Understanding how objects in nature move and deform is an essential part of any model of the world. Over the years, our community has developed countless models of dynamic objects, but most of these are specific to a particular object *type*, such as faces, hands, humans or quadrupeds (Blanz & Vetter, 1999; Romero et al., 2022; Loper et al., 2015; Zuffi et al., 2017). The few more general ones (Tang et al., 2022) do not make strong assumptions on the type of objects modelled, but are difficult to train due to the lack of suitable data (*e.g.*, aligned 3D meshes for (Tang et al., 2022)). None of these are good candidates for learning a 'foundation' model of object dynamics. Such a model should be able to express different types of object dynamics, such as part articulation, sliding of parts, and soft deformations. It must also be trainable on large quantities of Internet images and videos, so as to capture the diversity of objects that exist.

Recent video generators learned from millions of videos have been proposed as proxies of world models (Brooks et al., 2024). Such models should possess a general understanding of object dynamics. However, generating videos is insufficient: a useful dynamical model must be able to to make *predictions* about the motion of a given object, for example as the result of physical interactions. Inspired by DragAPart (Li et al., 2024c) and Yang et al. (2024), we thus consider learning a *conditional* video generator that makes prediction about the motion of objects in response to external stimuli. This generator takes as input a single image of an object and a set of *drags* which specify the motion of selected points of the object; it then outputs a physically plausible video of the object motion consistent with the drags (Fig. 1).

Several authors have already considered incorporating drag-like motion prompts in image or video generation (Blattmann et al., 2021; Chen et al., 2023; Pan et al., 2023; Yin et al., 2023; Li et al., 2024e; Wang et al., 2023; Shi et al., 2024; Mou et al., 2024b; Geng & Owens, 2024; Ling et al., 2024; Wu et al., 2024; Mou et al., 2024a; Li et al., 2024d). Many such works utilize techniques like

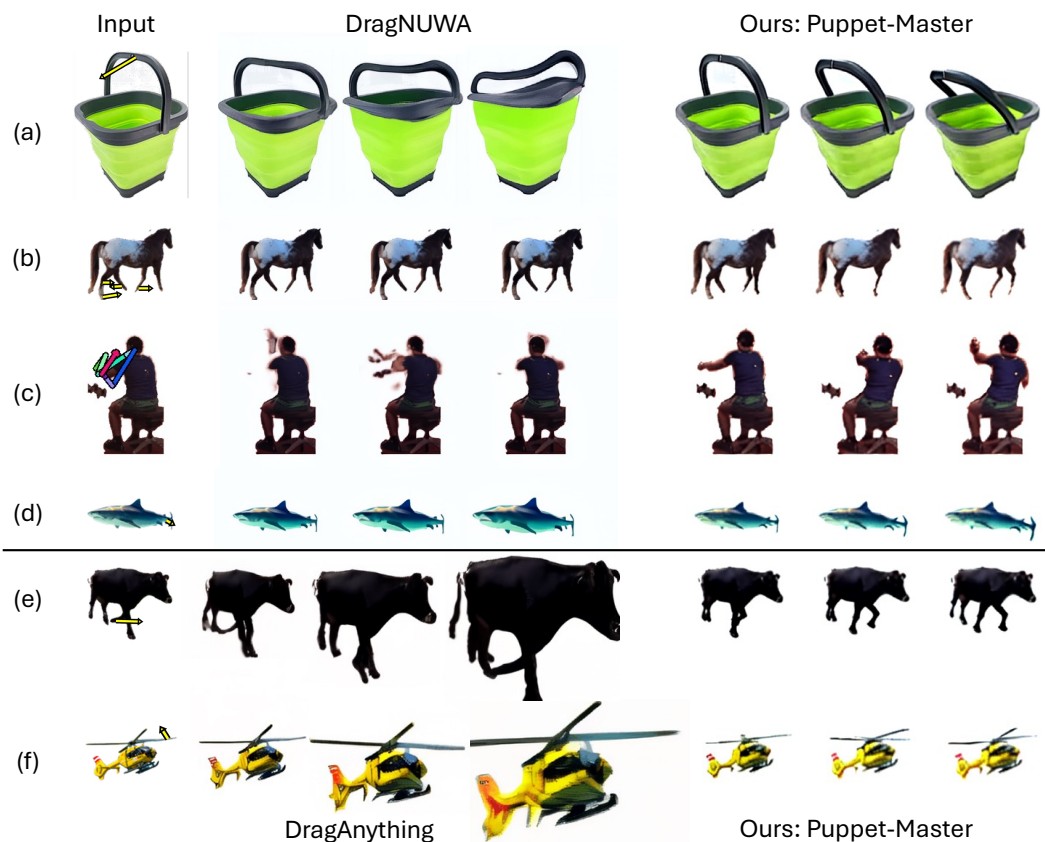

Figure 1: **Part-level dynamics *vs.* shifting or scaling an entire object.** PuppetMaster generates videos depicting *physically plausible part-level* motion, prompted by one or more drags (arrows).

ControlNet (Zhang et al., 2023) to inject motion control in a pre-trained generator. However, these models tend to respond to drags by shifting or scaling an entire object and fail to capture their internal dynamics, such as a drawer sliding out of a cabinet or a fish swinging its tail (Fig. 1). The challenge is encouraging generative models to synthesize such *internal, part-level* dynamics. While DragAPart has already considered *part-level* controllable generation, its results are limited for two reasons. First, the diversity of its training data is poor, as it primarily focuses on renderings of 3D furniture, instead of motion dynamics of various categories. Second, it starts from an image generator instead of a video generator. Consequently, it cannot benefit from the motion prior that a video generator may already contain, and can only predict the final state of the object, after the motion has occurred.

In this work, we thus explore the benefits of learning a motion model from a large-scale pre-trained video generator while also significantly scaling the necessary training data to larger, more diverse sources. In particular, we start from Stable Video Diffusion (SVD) (Blattmann et al., 2023a) and show how to re-purpose it for motion prediction. We make the following contributions.

First, we propose new conditioning modules to inject the dragging control into the video generation pipeline effectively. In particular, we find that *adaptive layer normalization* (Perez et al., 2018) is much more effective than the shift-based modulation proposed by Li et al. (2024c). We further observe that the cross-attention modules of the image-conditioned SVD model lack spatial awareness, and propose to add *drag tokens* to these modules for better conditioning. More importantly, we also address the degradation in appearance quality that often arises when fine-tuning video generators on out-of-distribution datasets by introducing *all-to-first* attention, where all generated frames attend the first one via varietal self-attention.AV: ? This design creates a shortcut that allows information to propagate from the conditioning frame to the other ones directly, significantly improving generation quality.

Our second contribution is to provide two datasets to learn part-level object motion. Both datasets comprise subsets of the 40k animated assets in Objaverse (Deitke et al., 2023). Objaverse animations vary in quality: some display realistic object dynamics, while others feature objects that (i) are static, (ii) exhibit simple translations, rotations, or scaling, or (iii) move in a physically implausible way. We introduce a systematic approach to curate the animations at scale. The resulting datasets, Objaverse-Animation and Objaverse-Animation-HQ, contain progressively fewer animations of higher quality. We show that Objaverse-Animation-HQ, which contains fewer but higher-quality animations, leads to a better model than Objaverse-Animation, demonstrating the effectiveness of the data curation.

With these new curated datasets, we train **PuppetMaster**, a video generative model that, given as input a single image of an object and corresponding drags, generates an animation of the object. These animations are *faithful to both the input image and the drags while containing physically plausible motions at the level of the individual object parts*. The same model works for a diverse set of object categories. Empirically, it outperforms prior works on multiple benchmarks. Notably, while our model is fine-tuned using only synthetic data, it generalizes well to real data, outperforming prior models that were fine-tuned on real videos. It does so in a *zero-shot* manner by generalizing to out-of-distribution, real-world data without further tuning.

## 2 RELATED WORK

**Generative models.** Recent advances in generative models, largely powered by diffusion models (Ho et al., 2020; Song & Ermon, 2019; Song et al., 2021), have enabled photo-realistic synthesis of images (Ramesh et al., 2021; Rombach et al., 2022; Saharia et al., 2022) and videos (Ho et al., 2022; Blattmann et al., 2023b; Girdhar et al., 2023; Blattmann et al., 2023a), and been extended to various other modalities (Tevet et al., 2022; Lei et al., 2023). The generation is mainly controlled by a text or image prompt. Recent works have explored ways to leverage these models' prior knowledge, via either score distillation sampling (Poole et al., 2023; Lin et al., 2023; Melas-Kyriazi et al., 2023; Jakab et al., 2024) or fine-tuning on specialized data for downstream applications, such as multi-view images for 3D asset generation (Liu et al., 2023; Li et al., 2024b; Melas-Kyriazi et al., 2024; Zheng & Vedaldi, 2024; Voleti et al., 2024; Gao et al., 2024).

**Video generation for motion.** Attempts to model object motion often resort to pre-defined shape models, *e.g.*, SMPL (Loper et al., 2015) for humans and SMAL (Zuffi et al., 2017) for quadrupeds, which are constrained to a single or only a few categories. Videos have been considered as a unified representation that can capture general object dynamics (Yang et al., 2024; Brooks et al., 2024). However, existing video generators pre-trained on Internet videos often suffer from incoherent or minimal motion. Researchers have considered explicitly controlling video generation with motion trajectories. Teng et al. (2023) extends the framework proposed by Pan et al. (2023) to videos. This method is training-free, relying on the motion prior captured by the pre-trained video generator, which is often not strong enough to produce high-quality videos. Hence, other works focus on training-based methods, which *learn* drag-based control using ad-hoc training data for this task. Early efforts such as Blattmann et al. (2021); Davtyan & Favaro (2024) train variational autoencoders or diffusion models to synthesize videos with objects in motion, conditioned on sparse motion trajectories sampled from optical flow. Li et al. (2024e) use a Fourier-based motion representation suitable for natural, oscillatory dynamics such as those of trees and candles, and generates motion for these categories with a diffusion model. DragNUWA (Yin et al., 2023) and others (Wang et al., 2023; Wu et al., 2024; Mou et al., 2024a; Li et al., 2024d) fine-tune pre-trained video generators on large-scale curated datasets, enabling drag-based control in open-domain video generation. However, these methods do *not* allow controlling motion at the level of object parts, as their training data entangles multiple factors, including camera viewpoint and object scaling and re-positioning, making it hard to obtain a model of part-level motion. Concurrent works leverage the motion prior captured by video generative models for the related 4D generation task (Liang et al., 2024; Zhang et al., 2024; Jiang et al., 2024; Xie et al., 2024). These models, however, lack the capability of explicit dragging control, which we tackle in this work.

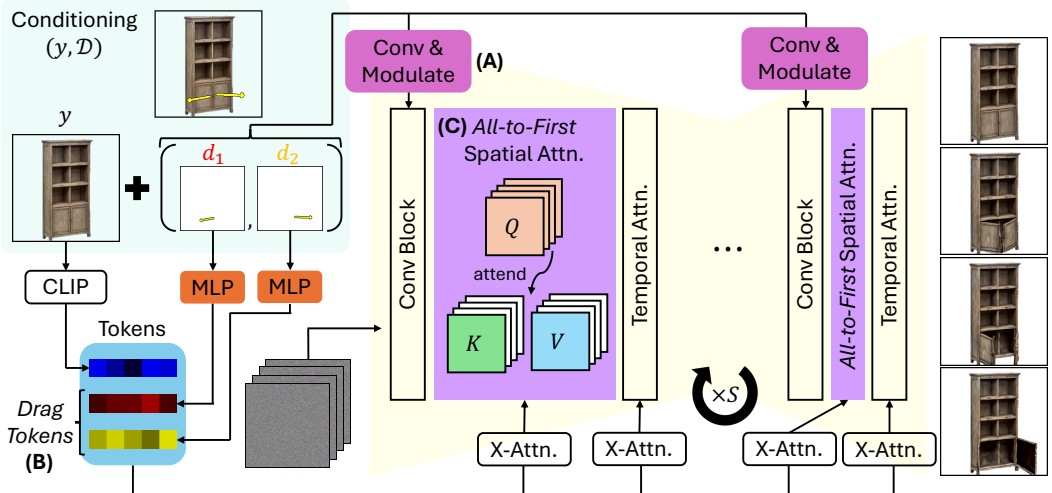

Figure 2: **Architectural Overview of PuppetMaster**. To enable precise drag conditioning, we first modify the original latent video diffusion architecture (Section 3.1) by (**A**) adding adaptive layer normalization modules to modulate the internal diffusion features and (**B**) adding cross attention with *drag tokens* (Section 3.2). Furthermore, to ensure high-quality appearance and background, we introduce (**C**) *all-to-first* spatial attention, a drop-in replacement for the spatial self-attention modules, where every video frame attends the first one (Section 3.3).

# 3 METHOD

Given the initial state of an object, represented by an image $y$, and one or more drags $\mathcal{D} = \{d_k\}_{k=1}^K$, our goal is to synthesize a video $\mathcal{X} = \{x_i\}_{i=1}^N$ sampled from the distribution $\mathcal{X} \sim \mathbb{P}(x_1, x_2, \ldots, x_N | y, \mathcal{D})$ where $N$ is the number of video frames. The distribution $\mathbb{P}$ should reflect physics and generate a *part-level* animation of the object that responds to the drags. To learn it, we capitalize on a large-scale pre-trained video generator, *i.e.*, Stable Video Diffusion (SVD, Section 3.1) (Blattmann et al., 2023a). Video generators have a general-purpose understanding of motion, acquired by pre-training on millions of Internet videos. This is key since there is only a limited amount of data representative of part-level object dynamics that can be used to train our model.

In this section, we show how to fine-tune such a pre-trained video generator to control the motion of objects at the level of it parts. There are two main challenges. First, the drag conditioning must be injected into the video generation pipeline to facilitate efficient learning and accurate and time-consistent motion control. This must be done without changing too much the internal pre-trained video representation. Second, naïvely fine-tuning a pre-trained video diffusion model can result in artefacts such as cluttered backgrounds (Li et al., 2024b). To address these challenges, in Section 3.2, we first introduce a novel mechanism to inject the drag condition $\mathcal{D}$ in the video diffusion model. Then, in Section 3.3, we improve the quality of the generated videos by introducing *all-to-first* attention mechanism, which reduces artefacts like the background clutter. While we build on SVD, these techniques should be easily portable to other video generators based on diffusion.

## 3.1 PRELIMINARIES: STABLE VIDEO DIFFUSION

SVD is an image-conditioned video generator based on diffusion, implementing a denoising process in latent space. It utilizes a variational autoencoder (VAE) $(E, D)$, where the encoder $E$ maps the video frames to the latent space, and the decoder $D$ reconstructs the video from the latent codes. During training, given a pair $(\mathcal{X} = x^{1:N}, y)$ formed by a video and the corresponding image prompt, one first obtains the latent code as $z_0^{1:N} = E(x^{1:N})$, and then adds to the latter Gaussian noise $\epsilon \sim \mathcal{N}(0, \boldsymbol{I})$, obtaining the progressively more noised codes

$$z_t^{1:N} = \sqrt{\bar{\alpha}_t} z_0^{1:N} + \sqrt{1 - \bar{\alpha}_t} \epsilon^{1:N}, \quad t = 1, \ldots, T. \tag{1}$$

This uses a pre-defined noising schedule $\bar{\alpha}_0 = 1, \ldots, \bar{\alpha}_T = 0$. The denoising network $\epsilon_\theta$ is trained to reverse this noising process by optimizing the objective function:

$$\min_\theta \mathbb{E}_{(x^{1:N}, y), t, \epsilon^{1:N} \sim \mathcal{N}(0, \boldsymbol{I})} \left[ \| \epsilon^{1:N} - \epsilon_\theta(z_t^{1:N}, t, y) \|_2^2 \right]. \tag{2}$$

Here, $\epsilon_\theta$ uses the same U-Net architecture of Blattmann et al. (2023b), inserting temporal convolution and temporal attention modules after the spatial modules used by Rombach et al. (2022). The image conditioning is achieved via (1) cross attention with the CLIP embedding of the reference frame $y$; and (2) concatenating the encoded reference image $E(y)$ channel-wise to $z_t^{1:N}$ as the input of the network. After $\epsilon_\theta$ is trained, the model generates a video $\hat{\mathcal{X}}$ prompted by $y$ via iterative denoising from pure Gaussian noise $z_T^{1:N} \sim \mathcal{N}(0, \boldsymbol{I})$, followed by VAE decoding $\hat{\mathcal{X}} = \hat{x}^{1:N} = D(z_0^{1:N})$.

## 3.2 ADDING DRAG CONTROL TO VIDEO DIFFUSION MODELS

Next, we show how to add the drags $\mathcal{D}$ as an additional input to the denoiser $\epsilon_\theta$ for motion control. This is achieved by introducing an encoding function for the drags $\mathcal{D}$ and by extending the SVD architecture to inject the resulting code into the network. The model is then fine-tuned using videos combined with corresponding drag prompts in the form of training triplets $(\mathcal{X}, y, \mathcal{D})$. We summarize the key components of the model below and refer the reader to Appendix A for more details.

**Drag encoding.** Let $\Omega$ be the spatial grid $\{1, \ldots, H\} \times \{1, \ldots, W\}$ where $H \times W$ is the resolution of the video. A *drag* $d_k$ is a tuple $(u_k, v_k^{1:N})$ specifying that the drag starts at location $u_k \in \Omega$ in the reference image $y$ and lands at locations $v_k^n \in \Omega$ in subsequent frames. To encode a set of drags $\mathcal{D} = \{d_k\}_{k=1}^K$, where $K \leq K_{\max} = 5$, we use the multi-resolution encoding of Li et al. (2024c). Each drag $d_k$[1] is fed to a hand-crafted encoding function $\mathrm{enc}(\cdot, s) : \Omega^N \mapsto \mathbb{R}^{N \times s \times s \times c}$, where $s$ is the desired encoding resolution. The encoding function captures the state of the drag in each frame; specifically, each slice $\mathrm{enc}(d_k, s)[n]$ encodes (1) the drag's starting location $u_k$ in the reference image, (2) its intermediate location $v_k^n$ in the $n$-th frame, and (3) its final location $v_k^N$ in the final frame. The $s \times s$ map $\mathrm{enc}(d_k, s)[n]$ is filled with values $-1$ except in correspondence of the 3 locations, where we store $u_k$, $v_k^n$ and $v_k^N$ respectively, utilizing $c = 6$ channels. Finally, we obtain the encoding $\mathcal{D}_{\mathrm{enc}}^s \in \mathbb{R}^{N \times s \times s \times c K_{\max}}$ of $\mathcal{D}$ by concatenating the encodings of the $K$ individual drags, filling extra channels with value $-1$ if $K < K_{\max}$. The encoding function is further detailed in Appendix A.

**Drag modulation.** The SVD denoiser comprises a sequence of U-Net blocks operating at different resolutions $s$. We inject the drag encoding $\mathcal{D}_{\mathrm{enc}}^s$ in each block, matching the block's resolution $s$. We do so via modulation using an adaptive normalization layer (Perez et al., 2018). Namely,

$$f_s \leftarrow f_s \otimes (\boldsymbol{1} + \gamma_s) + \beta_s, \tag{3}$$

where $f_s \in \mathbb{R}^{B \times N \times s \times s \times C}$ is the U-Net features of resolution $s$, and $\otimes$ denotes element-wise multiplication. $\gamma_s, \beta_s \in \mathbb{R}^{B \times N \times s \times s \times C}$ are the *scale* and *shift* terms regressed from the drag encoding $\mathcal{D}_{\mathrm{enc}}^s$. We use convolutional layers to embed $\mathcal{D}_{\mathrm{enc}}^s$ from the dimension $c K_{\max}$ to the target dimension $C$. We empirically find that this mechanism provides better conditioning than using only a single shift term with *no* scaling as in Li et al. (2024c) (ablated in Table 2).

**Drag tokens.** In addition to conditioning the network via drag modulation, we also do so via cross-attention by exploiting SVD's cross-attention modules. These modules attend a *single* key-value obtained from the CLIP (Radford et al., 2021) encoding of the reference image $y$. Thus, they degenerate to a global bias term with *no* spatial awareness (Sobol et al., 2024). In contrast, we concatenate to the CLIP token additional *drag tokens* so that cross-attention is non-trivial. We use multi-layer perceptrons (MLPs) to regress an additional key-value pair from *each* drag $d_k$. The MLPs take the origin $u_k$ and terminations $v_k^n$ and $v_k^N$ of $d_k$ along with the internal diffusion features sampled at these locations, which are shown to contain semantic information (Baranchuk et al., 2021), as inputs. Overall, the cross-attention modules have $1 + K_{\max}$ key-value pairs (1 is the original image CLIP embedding), with extra pairs set to 0 if $K < K_{\max}$.

---

[1]With a slight abuse of notation, we assume $d_k \in \Omega^N$, as $u_k = v_k^1$ and hence $v_k^{1:N} \in \Omega^N$ fully describes $d_k$.

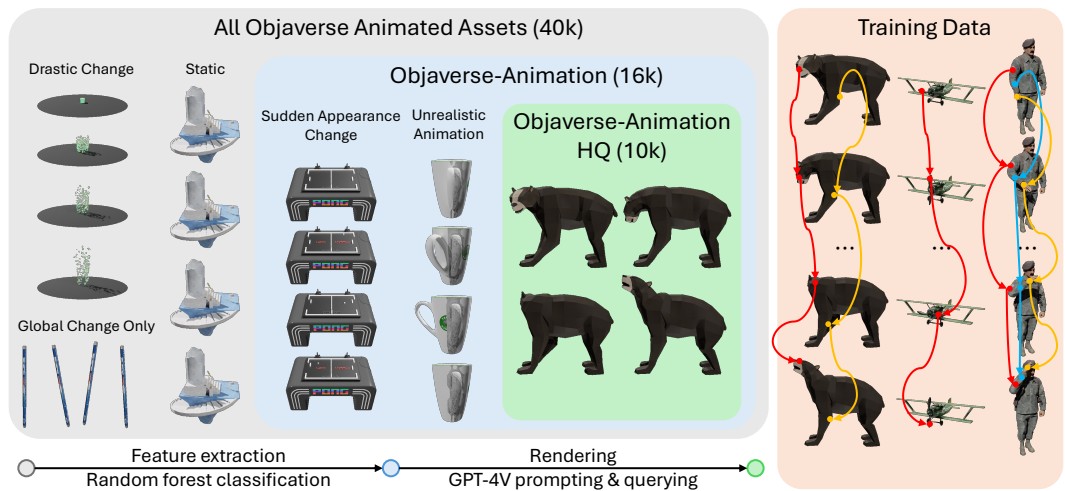

Figure 3: **Data Curation**. We propose two strategies to filter the animated assets in Objaverse, resulting in Objaverse-Animation (16k) and Objaverse-Animation-HQ (10k) of varying levels of curation, from which we construct the training data of PuppetMaster by sampling sparse motion trajectories and projecting them to 2D as drags.

### 3.3 ATTENTION WITH THE REFERENCE IMAGE COMES TO RESCUE

In preliminary experiments utilizing the Drag-a-Move (Li et al., 2024c) dataset, we noted that the generated videos tend to have cluttered/gray backgrounds. Instant3D (Li et al., 2024b) reported a similar problem when generating multiple views of a 3D object, which they addressed via careful noise initialization. VideoMV (Zuo et al., 2024) and Vivid-ZOO (Li et al., 2024a) directly constructed training videos with a gray background, which might help them offset a similar problem.

The problem is that SVD, which was trained on $576 \times 320$ videos, fails to generalize to very different resolutions, as shown by the failure of SVD to produce a reasonable video when prompted by a $256 \times 256$ image. Thus, fine-tuning SVD on $256 \times 256$ videos, as we do here, results in sub-optimal generations. However, we noticed that the first frame of each generated video is spared from the appearance degradation (Fig. 5), as the model learns to directly copy the reference image. Inspired by this, we introduce a *shortcut* from each noised frame to the first frame via attention. We call this *all-to-first* spatial attention, and shows that it almost entirely solves this problem.

**All-to-first spatial attention.** Previous works (Watson et al., 2023; Cao et al., 2023; Weng et al., 2023) have shown that attention between the noised branch and the reference branch improves the generation quality of image editing and novel view synthesis tasks. Here, we use *all-to-first* spatial attention where each noised frame to attend to the first (reference) frame. Inspired by Weng et al. (2023), we implement this attention by having each frame query the key and value of the first frame, changing all self-attention layers in the denoising U-Net. More specifically, denoting the query, key, and value tensors as $Q, K$ and $V \in \mathbb{R}^{B \times N \times s \times s \times C}$, we discard the key and value tensors of non-first frames, *i.e.*, $K[:, 1:]$ and $V[:, 1:]$, and compute the spatial attention $A_i$ of the $i$-th frame as follows:

$$A_i = \text{softmax}\left(\frac{\text{flat}\left(Q[:, \mathtt{i}]\right)\text{flat}\left(K[:, \mathtt{0}]\right)^T}{\sqrt{D}}\right)\text{flat}\left(V[:, \mathtt{0}]\right), \tag{4}$$

where $\text{flat}(\cdot): \mathbb{R}^{B \times s \times s \times C} \mapsto \mathbb{R}^{B \times L \times C}$ flattens the spatial dimensions to get $L = s \times s$ tokens for attention. The benefit is two-fold: first, this shortcut to the first frame allows subsequent frames to directly access non-degraded appearance details of the reference image. Second, combined with the proposed drag encoding (Section 3.2), which specifies, for *every* frame, the origin $u_k$ at the first frame, all-to-first attention enables the latent pixel containing the drag termination (*i.e.*, $v_k^n$) to more easily attend to the latent pixel containing the drag origin on the first frame, facilitating learning.

## 4 CURATING DATA TO LEARN PART-LEVEL OBJECT MOTION

For training, we require a video dataset that captures the motion of objects at the level of parts. Creating such a dataset in the real world means capturing a large number of videos of moving objects while controlling for camera and background motion. This is difficult to do for many categories (*e.g.*, animals) and unfeasible at scale. Li et al. (2024c) used instead renderings of synthetic 3D objects, and their corresponding part annotations, obtained from GAPartNet (Geng et al., 2023). Unfortunately, this dataset still requires to manually annotate and animate 3D object parts, which limits its scale. We instead turn to Objaverse (Deitke et al., 2023), a large-scale 3D dataset of 800k models created by 3D artists, among which 40k are animated. In this section, we introduce a pipeline to extract suitable training videos from these animated assets, together with corresponding drags $\mathcal{D}$.

**Identifying animations.** While Objaverse (Deitke et al., 2023) has 40k assets labeled as animated, not all animations are useful for our purposes (Fig. 3). Notably, some are "fake", with the objects remaining static throughout the sequence, while others feature drastic changes in the objects' positions or even their appearances. Therefore, our initial step is to filter out these unsuitable animations. To do so, we extract a sequence of aligned point clouds from each animated model and calculate several metrics for each sequence, including: (1) the dimensions and location of the bounding box encompassing the entire motion clip, (2) the size of the largest bounding box for the point cloud at any single timestamp and (3) the mean and maximal total displacement of all points throughout the sequence. Using these metrics, we fit a random forest classifier, which decides whether an animation should be included in the training videos or not, on a subset of Objaverse animations where the decision is manually labeled. The filtering excludes many assets that exhibit imperceptibly little or over-dramatic motions and results in a subset of 16k animations, which we dub Objaverse-Animation.

Further investigation reveals that this subset still contains assets whose motions are artificially conceived and therefore do not accurately mimic real-world dynamics (Fig. 3). To avoid such imaginary dynamics leaking into our synthesized videos, we employ the multi-modal understanding capability of GPT-4V (OpenAI, 2023) to assess the realism of each motion clip. Specifically, for each animated 3D asset in Objaverse-Animation, we fix the camera at the front view and render 4 images at timestamps corresponding to the 4 quarters of the animation and prompt GPT-4V to determine if the motion depicted is sufficiently realistic to qualify for the training videos. This filtering mechanism excludes another 6k animations, yielding a subset of 10k animations which we dub Objaverse-Animation-HQ.

**Sampling drags.** The goal of drag sampling is to produce a sparse set of drags $\mathcal{D} = \{d_k\}_{k=1}^K$ where each drag $d_k := (u_k, v_k^{1:N})$ tracks a point $u_k$ on the asset in pixel coordinates throughout the $N$ frames of rendered videos. To encourage the video generator to learn a meaningful motion prior, the set should ideally be both *minimal* and *sufficient*: each group of independently moving parts should have *one* and *only one* drag corresponding to its motion trajectory, similar to Drag-a-Move (Li et al., 2024c). For instance, there should be separate drags for different drawers of the same furniture, as their motions are independent, but not for a drawer and its handle, as in this case, the motion of one *implies* that of the other. However, Objaverse (Deitke et al., 2023) lacks the part-level annotation to enforce this property. To partially overcome this, we find that some Objaverse assets are constructed in a bottom-up manner, consisting of multiple sub-models that align well with semantic parts. For these assets, we sample one drag per sub-model; for the rest, we sample a random number of drags in total. For each drag, we first sample a 3D point on the visible part of the model (or sub-model) with probabilities proportional to the point's total displacement across $N$ frames and then project its ground-truth motion trajectory $p_1, \ldots, p_N \in \mathbb{R}^3$ to pixel space to obtain $d_k$. Once all $K$ drags are sampled, we apply a post-processing procedure to ensure that each pair of drags is sufficiently distinct, *i.e.*, for $i \neq j$, we randomly remove one of $d_i$ and $d_j$ if $\|v_i^{1:N} - v_j^{1:N}\|_2^2 \leq \delta$ where $\delta$ is a threshold we empirically set to $20N$ for $256 \times 256$ renderings.

## 5 EXPERIMENTS

PuppetMaster is trained on a combination of dataset: Drag-a-Move (Li et al., 2024c) and our new Objaverse-Animation-HQ (Section 4). We evaluate the performance of the final checkpoint on multiple benchmarks, including the test split of Drag-a-Move and real data from Human3.6M (Ionescu

Table 1: **Comparisons** with DragNUWA, DragAnything and DragAPart on the in-domain Drag-a-Move and out-of-domain Human3.6M datasets. The **best** method is bolded and second best underlined. Our model has *not* been trained on the Human3.6M dataset, or any real video datasets.

| Method | Video | Drag-a-Move | | | | | Human3.6M | | | |
|---|---|---|---|---|---|---|---|---|---|---|
| | | PSNR↑ | SSIM↑ | LPIPS↓ | FVD↓ | flow error↓ | PSNR↑ | SSIM↑ | LPIPS↓ | FVD↓ |
| DragNUWA | ✓ | 20.09 | 0.874 | 0.172 | 281.49 | 17.55 / 15.41 | 17.52 | **0.878** | 0.158 | 466.91 |
| DragAnything | ✓ | 16.71 | 0.799 | 0.296 | 468.46 | 16.09 / 23.21 | 13.29 | 0.767 | 0.305 | 768.63 |
| DragAPart | | | | | | | | | | |
| *— Original* | ✗ | 23.41 | 0.925 | 0.085 | 180.27 | 14.17 / 3.71 | 15.14 | 0.852 | 0.197 | 683.40 |
| *— Re-Trained* | ✗ | 23.78 | **0.927** | **0.082** | **189.10** | 14.34 / 3.73 | 15.25 | 0.860 | 0.188 | 549.64 |
| PuppetMaster | ✓ | **24.41** | **0.927** | 0.085 | 246.99 | **12.21 / 3.53** | **17.59** | 0.872 | **0.155** | **454.76** |

et al., 2014), Amazon-Berkeley Objects (Collins et al., 2022), Fauna Dataset (Wu et al., 2023; Li et al., 2024f), and CC-licensed web images in a *zero-shot* manner (i.e., without tuning on real data), demonstrating qualitative and quantitative improvements over prior works and excellent generalization to real cases (Section 5.1). The design choices that led to PuppetMaster are ablated and discussed further in Section 5.2. In Appendix B.2, we show the effectiveness of our data curation strategy (Section 4). We refer the reader to Appendix C for the implementation details.

## 5.1 MAIN RESULTS

**Quantitative comparison.** In Table 1, we compare PuppetMaster on the task of drag-controlled video generation to DragNUWA (Yin et al., 2023) and DragAnything (Wu et al., 2024), two *video* generators trained for the same task using real data. On Drag-a-Move, where the goal is to control motion at the level of parts rather than whole objects, PuppetMaster outperforms both methods on all standard metrics, including PSNR, SSIM, LPIPS, and FVD, by a significant margin.

Additionally, to better test the ability of models to capture part-level dynamics accurately, we introduce a flow-based metric dubbed *flow error*. We first track the points on the object throughout the generated and ground-truth videos using CoTracker (Karaev et al., 2024), and then compute flow error as the root mean square error (RMSE) between corresponding trajectories, and report it in Table 1. The first value (before the slash) is averaged among the origins of all conditioning drags only, *i.e.*, $\{u_k\}_{k=1}^{K}$, while the second value (after the slash) is averaged among all foreground points. While PuppetMaster has lower values on both, it obtains a *significantly* smaller value when the error is averaged among all foreground points. This indicates that PuppetMaster captures part-level dynamics better; for example, the parts that do not *have* to move based on the specified input drags do not, which generally matches the ground truth and reduces the overall error. By contrast, DragNUWA and DragAnything always move the whole object, so many points incur large errors.

To assess the cross-domain generalizability, we evaluate PuppetMaster on an unseen dataset captured in the real world (*i.e.*, Human3.6M). On this out-of-domain test set, PuppetMaster outperforms prior models on most metrics, despite not being fine-tuned on any real videos. For completeness, we also include the metrics of DragAPart (Li et al., 2024c), a drag-conditioned *image* generator. The original DragAPart was trained on Drag-a-Move only. For fairness, we fine-tune it from Stable Diffusion (Rombach et al., 2022) with the identical data setting as PuppetMaster, and evaluate the performance of both checkpoints (*Original*[2] and *Re-Trained* in Table 1). The videos are obtained from $N$ independently generated frames conditioned on gradually extending drags. While its samples exhibit high visual quality in individual frames, they lack temporal smoothness, characterized by abrupt transitions and discontinuities in movement, resulting in a larger flow error[3] (Fig. 7a in sup. mat.). This justifies starting from a video generator to improve temporal consistency. Furthermore, DragAPart fails to generalize to out-of-domain cases (*e.g.*, Fig. 7b in sup. mat. and Table 1).

---

[2] *Original* is not ranked as it is trained on single-category data only and hence not an open-domain generator.

[3] FVD is *not* an informative metric for motion quality. Prior works (Ge et al., 2024; Watson et al., 2024) noted that FVD is biased towards the quality of individual frames and does *not* sufficiently account for motion. Good FVD scores can still be obtained with static videos or videos with severe temporal corruption.

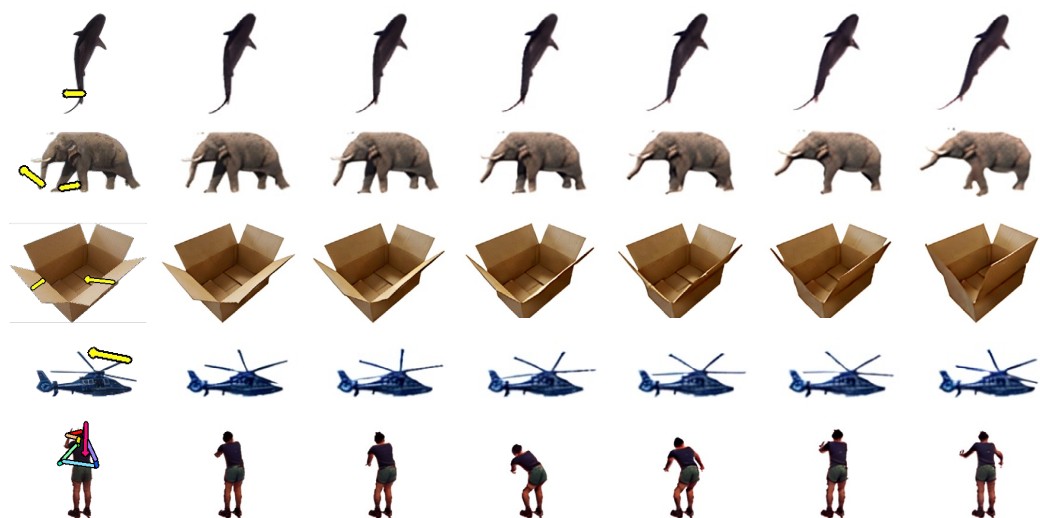

Figure 4: **Qualitative Results** on *real-world* cases spanning diverse categories.

**Qualitative comparison.** We show samples generated by PuppetMaster and prior models side by side in Fig. 1. The dynamics generated by PuppetMaster are physically plausible and faithful to the input image and drags. By contrast, the videos generated by DragNUWA and DragAnything scale (d, e, f) or shift (b) the object as a whole at best, or even show distorted motion (a, c). Even though PuppetMaster is fine-tuned solely on renderings of synthetic 3D models, it *does* generalize to real cases, and is capable of preserving fine-grained texture details.

**Qualitative results on real data.** In Fig. 4, we show more real examples generated by PuppetMaster. The synthesized videos exhibit realistic dynamics that are typical of the underlying categories, including humans, animals, and several man-made categories.

Table 2: **Ablation** studies of various model components. In addition to the standard metrics, we report a flow-based metric dubbed *flow error*. A lower flow error indicates the generated videos follow the drag control better. We also manually count the frequency of generated videos whose motion directions are opposite to the intention of their drag inputs. Here, $\geq$ indicates there are video samples whose motion directions are hard to distinguish. When ablating attention with the reference image, we use $\mathbb{C}$ as the base drag conditioning architecture.

| Setting | | PSNR↑ | SSIM↑ | LPIPS↓ | FVD↓ | flow error↓ | % wrong dir.↓ |
|---|---|---|---|---|---|---|---|
| **Drag conditioning** | | | | | | | |
| $\mathbb{A}$ | Shift only w/o end loc. | 13.23 | 0.816 | 0.446 | 975.16 | 15.60 px | $\geq 5$ |
| $\mathbb{B}$ | Shift+scale w/o end loc. | 22.98 | 0.917 | 0.093 | 223.20 | **9.33** px | 4 |
| $\mathbb{C}$ | Shift+scale w/ end loc. | 23.67 | 0.926 | 0.080 | 205.40 | 10.48 px | 4 |
| $\mathbb{D}$ | $\mathbb{C}$ + x-attn. w/ drag tok. | **24.00** | **0.929** | **0.069** | **170.43** | 9.80 px | **1** |
| **Attn. w/ ref. image** | | | | | | | |
| No attn. | | 11.96 | 0.771 | 0.391 | 823.00 | 12.35 px | $\geq 3$ |
| Attn. w/ static ref. video | | 17.51 | 0.874 | 0.233 | 483.18 | 13.57 px | $\geq 8$ |
| *All-to-first* attn. | | **23.67** | **0.926** | **0.080** | **205.40** | **10.48** px | 4 |

## 5.2 ABLATIONS

We conduct several ablation studies to analyze the introduced components of PuppetMaster. For each design choice, we train a model using the training split of the Drag-a-Move dataset with batch size 8 for 30k iterations and evaluate on 100 videos from its test split without classifier-free guidance (Ho & Salimans, 2022). Results are shown in Table 2 and Fig. 5 and discussed in detail next.

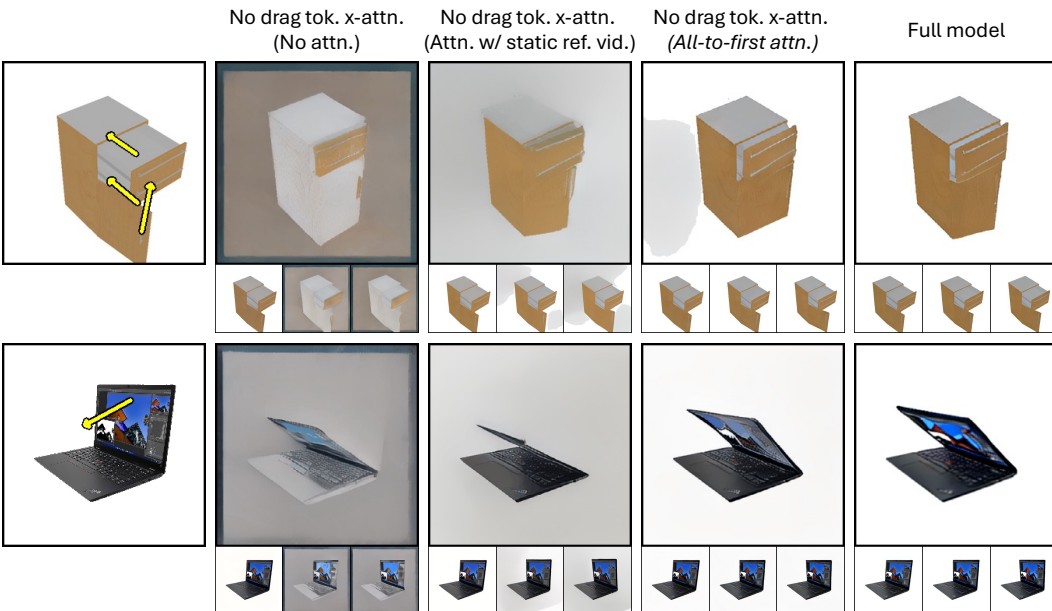

Figure 5: **Visualization** of samples generated by different model designs, where we show the last frame and the first 3 frames. While all designs produce nearly perfect first frames, our proposed *all-to-first* attention module significantly enhances sample quality. Without this module, the generated samples often exhibit sub-optimal appearances and backgrounds. The cross-attention module with drag tokens further improves the appearance details.

**Drag conditioning.** Table 2 compares PuppetMaster with several variants of conditioning mechanisms (Section 3.2). Adaptive normalization layers ($\mathbb{A}$ *vs.* $\mathbb{B}$) significantly improve the both appearance quality (PSNR by about 9 points) and motion consistency (flow error by about 6 points) of generated videos. This highlights the effectiveness of the new module in enhancing the visual fidelity and temporal coherence of the generated videos. Additionally, we perform an ablation study on the impact of drag encoding with final termination location $v_k^N$ ($\mathbb{B}$ *vs.* $\mathbb{C}$). This also proves beneficial for producing the final motion state of objects. Notably, by combining these (*i.e.*, row $\mathbb{D}$), the model achieves a negligible rate of generated samples with incorrect motion directions (see Table 2).

**Attention with the reference image.** An evaluation of our proposed *all-to-first* attention is shown in Table 2 and Fig. 5. We find that *all-to-first attention* (Section 3.3) is essential for high generation quality. We also compare *all-to-first* attention with an alternative implementation strategy inspired by the X-UNet design by Watson et al. (2023), where we pass a static video consisting of the reference image copied $N$ times to the same network architecture and implement cross attention between the clean (static) reference video branch and the noised video branch. The latter strategy performs worse. We hypothesize that this is due to the distribution drift between the two branches, which forces the optimization to modify the pre-trained SVD's internal representations too much.

## 6 CONCLUSION

We have introduced PuppetMaster, a video generator that allows to control the motion of objects at the level of their parts via one or more drags. Compared to related works, PuppetMaster incorporates several architectural innovations, such as the adaptive layer normalization modules, the cross-attention modules with drag tokens, and the all-to-first spatial attention modules. Ablation demonstrates the efficacy of these contributions. PuppetMaster is trained on Objaverse-Animation-HQ, a new curated dataset of part-level object animations, that we also contributed. PuppetMaster achieves state-of-the-art performance on several benchmarks and strong *zero-shot* generalization to real-world cases. Most importantly, it demonstrates the viability of using video generators as proxies to learn a foundation model of the internal dynamics of objects.

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
