# OpenReview forum: "PuppetMaster: Scaling Interactive Video Generation as a Motion Prior for Part-Level Dynamics"
_ICLR.cc/2025/Conference — Submitted to ICLR 2025_

### Official Review · Reviewer_H6mH · 2024-11-02

**Soundness:** 2
**Presentation:** 2
**Contribution:** 2
**Rating:** 5
**Confidence:** 3

**Summary:**

This paper addresses the problem of generating videos controlled by drag. The key idea is to take the pre-trained stable video diffusion and fine-tune it by conditioning it on drag control. They used all-to-first attention to improve the consistency between the reference images and video frames. A dataset with paired drag input and output videos is collected while building upon the Objaverse animation dataset.

**Strengths:**

First, the collected dataset could be helpful for further study in this direction.
Second, compared with previous works such as DragAPart, which used an image generation model, it makes more sense to use pre-trained video diffusion to generate temporal consistent videos.

**Weaknesses:**

First, some descriptions might be confusing to understand, making it difficult to follow. For example, in line 113, "The resulting datasets, Objaverse- Animation and Objaverse-Animation-HQ, contain progressively fewer animations of higher quality." what does it mean by progressively fewer animations?
From Lines 197-205, they mentioned there are two main challenges. However, it seems that both challenges are about injecting drag control into the pre-trained video diffusion.

Second, to my understanding, the problem itself is not quite properly defined. Given the drag input in the image space only, the input control itself is ambiguous. For example, whether the drag is supposed to zoom-in the object or manipulate part of the object is ambiguous. From my understanding, it also depends on the training set. In this case, the training is severely overfitted to the training set.

Third, the technical contributions are not clear here. Except for collecting a dataset, the technical novelty is limited. For example, the drag encoder seems to be very similar to that proposed in DragAPart. The all-to-first attention(replacing K, V in the attention) is also not new in the context of video generation or image generation, such as Consistent123.

**Questions:**

1) What is the difference between the drag encoder and that in the DragAPart paper?
2) Why not demonstrate the generated videos of DragAPart?
3) What are the failure cases?

---

> ### Author Response · Authors · 2024-11-16
> **Thank you & Responses to your Concerns and Questions**
>
> Thank you for your time devoted to reviewing our work and your feedback. We answered some concerns shared by fellow reviewers above. Here, we answer your remaining questions:
>
> ## Weakness 1
> “Progressively fewer animations” means **Objaverse-Animation-HQ goes through an extra filtering step** (L347-354), and hence **contains fewer but higher-quality animations**.
>
> We would like to think of the second challenge as an **optimization challenge**, i.e., how to facilitate learning so that the model doesn’t get stuck in local optimum. We then propose architectural changes (i.e., All-to-First Attention) to resolve this. That being said, it’s not wrong at all if you would like to think of them as a single challenge under the umbrella of “effectively injecting drag control”. We will provide this alternative interpretation in the text in the revision.
>
> ## Weakness 2
> In this work, our goal is to **synthesize internal, part-level motion, as opposed to shifting or scaling an entire object** (L86-90). We stated this again in L191-192: “**generate a part-level animation of the object**”. You are absolutely correct that **high-quality training data is the key to achieve this**. To this end, we dedicated enormous efforts into data curation to learn motion where object parts are manipulated (Sec. 4).
>
> We believe what you pointed out is a really interesting direction! We have had many internal discussions about learning a model that can interpret whether the drag is intended for part manipulation, object zooming, or object moving. An illustration from our discussion is provided in `future-work.png` in the revised Sup. Mat. However, this is beyond the scope of our submission.
>
> ## Weakness 3
> Please refer to the **Novelty of the All-to-First Attention** section of the shared thread.
>
> ## Q1
> There are **3 key differences**:
>
> (1) For each video frame $n$, in addition to encoding the starting point $v^1$ and the ending point at the current frame $v^n$, we also encode the final drag termination point $v^N$ (L241-246).
>
> (2) We use modulation instead of a shift-only approach proposed by DragAPart to modify the internal diffusion features (L248-258).
>
> (3) We introduce the use of drag tokens to make the cross attention modules in SVD spatial-aware (L259-269).
>
> ## Q2
> We **provide 8 comparison examples in `video-results/comparison-DragAPart.gif` in the revised Sup. Mat.** We agree that including them could better visualize our model’s improved temporal consistency.
>
> ## Q3
> The model might generate videos of a higher level of artifacts if the input image features an object rarely represented in the training data, such as **non-articulated objects like balls or phones**. A few examples are provided in `video-results/non-articulated.gif` in the revised Sup. Mat.

---

### Official Review · Reviewer_tFyM · 2024-11-03

**Soundness:** 3
**Presentation:** 2
**Contribution:** 2
**Rating:** 5
**Confidence:** 3

**Summary:**

The paper presents PuppetMaster, a conditional 2D video generation model that takes an reference object image and some conditional input drags then synthesizes a video with part-level object dynamics. The authors finetunes an off-the-shelf video diffusion model and extend it to encode input conditional drags. Additionally, they propose all-to-first attention method to remove the artifacts when generating out-of-domain video and improve the quality. They also made their own dataset, Objaverse-Animation-HQ, with data curation from Objaverse dataset. The authors argued that the proposed method generalizes well on real images and outperforms in real-world benchmarks in a zero-shot manner.

**Strengths:**

1. The motivation of the paper to generate object animation given a reference object image and user-controllable drag condition seems practical. Compared to the existing methods that generate a static image of the final-state of objects, the proposed method can produce fine-grained object motion as a video.

2. The numerical experiments are conducted in rigorous validation by retraining the comparable method (DragAPart) with an identical data setting as the proposed method. To validate the accuracy of part-level dynamics, the authors introduce the 'flow error' metric by computing errors between predicted and ground-truth point trajectories.

**Weaknesses:**

1. Insufficient generation quality: Although the authors leverage and finetune the video foundational model (Stable Video Diffusion), the generated video quality is not sufficient to determine that the learned model well-captures object dynamics. In the video results provided by the authors in supplementary materials, the generated results do not preserve the ground-truth object's appearance details, e.g., the generated object colors are changed in every frame (flickering effect). The reviewer wonders if the authors consider these low temporal smoothness problems and make any solution in the proposed method.

2. Limited qualitative result: The reviewer thinks that the authors should provide more qualitative results compared with DragAPart. Although DragAPart is a method that generates a static image, the video results can be obtained by dense sampling on the trajectory and defining fine-grained drag conditions. The reviewer found two samples in the supplementary document, but the reviewer thinks that more diverse object samples should be reported in the main paper.

**Questions:**

1. In Sec. 5.1, the authors argue that the proposed model obtains smaller errors when averaging all foreground points (L412). However, the reviewer thinks that the flow error is smaller with the averaged conditioning points. It would be helpful if the authors could provide more explanation and analysis of the results.

2. In Table 2, the authors report the ablation study results. In models A and B, the reviewer wonders why the authors do experiments without an end location of the drags. Is it possible to encode input drags without an end point? It would be great if the authors give more explanation of baseline models in ablation studies.

3. Impact of drag modulation with scale: The authors argue that utilizing the adaptive normalization layer helps to improve the generation quality. The reviewer needs more explanations about the authors' design choices.

---

> ### Author Response · Authors · 2024-11-16
> **Thank you & Responses to your Concerns and Questions**
>
> Thank you for your time devoted to reviewing our work and your feedback. We answered some concerns shared by fellow reviewers above. Here, we answer your remaining questions:
>
> ## Weakness 1
> We **further provide 6 examples in `video-results/additional-result.gif` in the revised Sup. Mat. to demonstrate our model’s performance on a wide range of objects**. We include 3 samples generated with the same drag prompt to show its ability to generate reasonable yet diverse part-level motion videos (second row).
> Regarding the slight appearance flickering, we hypothesize the culprit is the pre-trained SVD model. **The videos generated by SVD usually feature this appearance flickering** (e.g., the [rocket here](https://cdn-lfs.hf.co/datasets/huggingface/documentation-images/b2e217bf68d8abd676be5a99276e2422b2a850070c1b84bedf4bbf2784fd0b58?response-content-disposition=inline%3B+filename*%3DUTF-8%27%27output_rocket_with_conditions.gif%3B+filename%3D%22output_rocket_with_conditions.gif%22%3B&response-content-type=image%2Fgif&Expires=1731953179&Policy=eyJTdGF0ZW1lbnQiOlt7IkNvbmRpdGlvbiI6eyJEYXRlTGVzc1RoYW4iOnsiQVdTOkVwb2NoVGltZSI6MTczMTk1MzE3OX19LCJSZXNvdXJjZSI6Imh0dHBzOi8vY2RuLWxmcy5oZi5jby9kYXRhc2V0cy9odWdnaW5nZmFjZS9kb2N1bWVudGF0aW9uLWltYWdlcy9iMmUyMTdiZjY4ZDhhYmQ2NzZiZTVhOTkyNzZlMjQyMmIyYTg1MDA3MGMxYjg0YmVkZjRiYmYyNzg0ZmQwYjU4P3Jlc3BvbnNlLWNvbnRlbnQtZGlzcG9zaXRpb249KiZyZXNwb25zZS1jb250ZW50LXR5cGU9KiJ9XX0_&Signature=CbzInfUXR%7Eh49ZuLJyVxw2x5Z-uQJCNgzHkW7YkEUDYoIvx2mIr8FzLPTnoyzlkZGIOVic21fKnhOe0Eciz0fsZfboi%7ENtjJCmP3FynIlx2pq4cO7oyu98%7EpP9mLEEEIT2YwpqZdZ78ihvYpEJXtMgVbiRdrX3PVCpHPvBstB4TYFRFmGG1IrbdJKG7byaA-gysrzkzTFzgzkKCJyPj40GQF5ad4C6XBZLQRl3bIctQQJyJKgWEyrT5wx-rpqObfn9bBa5K%7EutPC9c3t041Zq01XodKaY2jpuvrvUzDUtE9X2P96Tj4hKLEru45QhnO%7Eem4jwwRW0mYhP9ZJsU%7Epow__&Key-Pair-Id=K3RPWS32NSSJCE)). Fine-tuning from a better video generator could potentially resolve this problem.
>
> ## Weakness 2
> Thank you for the suggestion! We **provide 8 comparison examples in `video-results/comparison-DragAPart.gif` in the revised Sup. Mat.** We will make room to provide a few such examples in the main paper.
>
> ## Q1
> Thank you for reading our paper so carefully! To clarify, the number **before** the slash (**12.21** for PuppetMaster) is averaged among conditioning drag points (L410), while the number **after** the slash (**3.53** for PuppetMaster) is averaged among all foreground points (L411). So the **flow error is smaller when averaged with all foreground points**. This is because PuppetMaster models part-level motion – the parts that do not move with the drag (e.g. the 2 other covers of the box in the 3rd row of Figure 4) remain static throughout the video. The **points on those static parts have almost 0 flow error**, so when included (when flow error is averaged among all foreground points), they make the flow error **lower**. We explained this in L413-416; we will make the explanation clearer in the revision.
>
> ## Q2
> Here, the **end location refers to the termination point of the entire drag**. Recall in line 237-238 we defined each drag as $v^{1:N}$. For each video frame $n$, the encoding always encodes the starting point $v^1$ and the ending point at the current frame $v^n$. “w/ end loc” refers to the fact that the encoding further encodes the final termination point $v^N$. We empirically found that adding $v^N$ facilitates learning and yields better generation quality. This was specified in L241-246; we will make it clearer in the Experiment section.
>
> ## Q3
> The **modulation is implemented as in Eq (3)**. We regress the shift and scale term from the drag encoding with a few convolutional layers (L254-256) to modulate the diffusion hidden features. This design follows from DragAPart [1], where they only use a shift term, i.e., $f_s\leftarrow f_s + \beta_s$. We empirically found that **adding the scale term yields significantly better generation quality** (Table 2 design A vs. B).
>
> [1] DragAPart: Learning a Part-Level Motion Prior for Articulated Objects, Li et al.

---

### Official Review · Reviewer_W1d9 · 2024-11-04

**Soundness:** 3
**Presentation:** 3
**Contribution:** 3
**Rating:** 6
**Confidence:** 4

**Summary:**

The authors present a framework called PuppetMaster for generating videos with part level object motion from a single image. The approach uses "drag" inputs to control fine grained parts of the generated video. This approach is trained on the Objaverse-Animation-HQ dataset that contains videos with part level motion annotation. State of the art quantitative performance is demonstrated on the videos generated for various classes (animals, humans, objects).

**Strengths:**

1. **Clarity**: The paper is well written with adequate attention to detail. All the design components are explained in detail to aid in reproducibility.
2. **Result Quality**: The generated videos show improved quality over baselines with similar drag controls.
3. **Comparison**: Both quantitative and qualitative performance is provided against a number of baselines. Additionally, out-of-domain performance on the Human3.6 datasets shows the efficacy of the approach.
4. **Ablations**: The provided ablations highlight the need for the all-to-first attention and the right parameterization needed to represent the drag constraints.

**Weaknesses:**

1. **Claims of Novelty**: The authors claim the "all-to-first" attention. Although not previously demonstrated in its exact form. The mechanism is essentially a variant of cross attention looking at KV pair from just the first frame. To that the claim that authors present a "novel all-to-first attention can be softened. For instance, one can highlight the finding of all-to-first attention while emphasizing that the variant of cross-attention that always looks at first frames is better than no attention at generating videos with fine grained part control.

**Questions:**

1. What is maximum number of frames that can be generated ? Can the all-to-first attention be replaced with a full 3D attention for better performance?
2. How is the performance of the framework on non articulated objects? In particular, providing a video example of a non articulated object along with arbitrary drags is helpful to demonstrate the framework's default behavior.
3. What happens when the drag condition input to the framework is zero? Does the network generate a video with arbitrary motion or just a static image? Showing an example of this setting would be helpful to understand "unconditional" synthesis  from the network.

---

> ### Author Response · Authors · 2024-11-16
> **Thank you & Responses to your Concerns and Questions**
>
> Thank you for your time devoted to reviewing our work and your feedback. We answered some concerns shared by fellow reviewers above. Here, we answer your remaining questions:
>
> ## Weakness
> Please refer to the Novelty of the All-to-First Attention section of the shared thread.
>
> ## Q1
> The model generates **14 frames per video**.
>
> **Yes**, All-to-First Attention can be replaced by full 3D attention. In fact, a recent work CAT3D [1] claims full 3D attention is essential for the generation quality. However, **full 3D attention is compute intensive**, as the compute (time and memory) required in attention goes up quadratically with the sequence length. CAT3D does **inference** on **16 A100 GPUs**, whereas our model was **trained** entirely on **a single A6000 GPU**.
>
> ## Q2
> We provide 3 examples of non-articulated objects in `video-results/non-articulated.gif` in the revised Sup. Mat. The videos do show some motion towards the intended directions. However, since these cases were not seen during training, the generated videos can have artifacts (e.g., the iPhone and the tree).
>
> ## Q3
> The architecture of our model is designed so that **having a zero-length drag** is different from **having no drags at all**. The former specifies that no motion is wanted for that point, so the generated video features minimal motion. The latter specifies no motion control, thus an arbitrary motion is generated. An example distinguishing the two is provided in `video-results/corner-cases.gif` in the revised Sup. Mat. Such corner cases were rarely sampled during training, so the generated videos can exhibit a slightly higher level of artifacts.
>
> [1] CAT3D: Create Anything in 3D with Multi-View Diffusion Models, Gao et al.

---

### Author Response · Authors · 2024-11-16
**Author Rebuttal: Responses to Shared Concerns & Questions**

We would like to thank all three fellow reviewers for your time devoted to providing valuable feedback to our work. We are encouraged by the positive comments: “**well written** (W1d9)”, “**improved quality** (W1d9)”, **rigorous validation and thorough comparison** (W1d9, tFyM) and the **recognition of our dataset contribution** (H6mH). We address individual questions in separate threads, but first highlight some shared concerns.

## Comparison with DragAPart (tFyM, H6mH)
We provided two examples in the supplementary material (Sup. Mat.). We agree that providing more examples, especially as videos, could better visualize our model’s improved temporal consistency. **We provide 8 examples in `video-results/comparison-DragAPart.gif` in the revised Sup. Mat.**

## Novelty of the All-to-First Attention (W1d9, H6mH)
The idea of using cross attention in image/video generation is not new. As pointed out by H6mH, the closest to our proposed All-to-First Attention is Consistent123 [1], where they attend each noised novel view to the key and value obtained from the conditioning view for the novel view synthesis (NVS) problem. Our work is highly inspired by [1] (we stated this explicitly in L307-308). **To the best of our knowledge, we are the first to introduce this cross-frame spatial attention mechanism in the domain of video generation.**

We would also like to emphasize that **All-to-First Attention is not our only contribution**. Generating intrinsic, part-level object motion prompted by drags is non-trivial, complicated further by the lack of high-quality data. In addition to the introduction of All-to-First Attention, (1) we **curate a large-scale high-quality 4D dataset, Objaverse-Animation-HQ**. Enormous efforts were dedicated to curating this dataset, from classifier training and GPT-4V prompting to large-scale rendering. We believe this dataset can significantly benefit the community. (2) We **introduce drag tokens to make the cross attention modules in SVD spatial-aware**. We believe this design can inspire future research in conditional image/video generation.

Thank you again for your inputs! We are more than willing to discuss further if you have follow-up questions regarding our work.

References:
[1] Consistent123: Improve Consistency for One Image to 3D Object Synthesis, Weng et al.

---

### Meta-Review · Area_Chair_Gd93 · 2024-12-19

**Metareview:**

The paper presents an approach to control articulated objects by "dragging" parts of the objects. The reviews place the paper slightly below the borderline. While the reviewers acknowledge the clarify of the manuscript, the practicality of the problem, and extensiveness of numerical experiments, they also list several issues and questions. The reviewers suggest that the quality of generations is expected to be better. The AC agrees with this statement. AC checked the supplement and found the quality insufficient. It was not clear whether it's the gif encoding that introduces color artifacts or the backbone video model. One of the reviewers mentioned that too. In general AC believes that choosing gif format in the supplement doesn't do a good service to the manuscript. Besides, the supplement just lacks enough comparisons. Reviewers mentioned that too, and additional comparisons were added during the discussion, however, AC believes that 8 comparisons is not enough.

In summary, the paper didn't get enough support from reviewers. That's due to quality, certain concerns with novelty and technical contribution, and the amount of visual results provided.

**Additional Comments On Reviewer Discussion:**

The authors tried to address the concerns during the discussion period. However, the scores didn't go up.

---

### Decision · Program_Chairs · 2025-01-22

Reject